# Senolytic Flavonoids Enhance Type-I and Type-II Cell Death in Human Radioresistant Colon Cancer Cells through AMPK/MAPK Pathway

**DOI:** 10.3390/cancers15092660

**Published:** 2023-05-08

**Authors:** Maria Russo, Stefania Moccia, Diomira Luongo, Gian Luigi Russo

**Affiliations:** Institute of Food Sciences, National Research Council, 83100 Avellino, Italy; stefania.moccia@isa.cnr.it (S.M.); diomira.luongo@isa.cnr.it (D.L.); glrusso@isa.cnr.it (G.L.R.)

**Keywords:** radioresistance, autophagy, SASP, natural senolytics, AMPK, ERKs kinases

## Abstract

**Simple Summary:**

The role of autophagy and senescence in cancer resistance to ionizing radiation as a response to genotoxic stress is still only partially explored. The flavonoids quercetin and fisetin have previously been shown to sensitize cancer cells resistant to radiotherapy by targeting p16^INK4^ and p21^Kip1^. Here, we examined their ability to modulate autophagy and senescence-associated inflammatory markers after irradiation in radioresistant cells. Quercetin or fisetin, in association with ionizing radiation, significantly activated AMPK and decreased ERK kinase activity, which was linked to autophagic stress response and apoptosis induction. In simple words, on one side, the combined treatment favored the induction of autophagy and senescence by activating AMPK; on the other side, it lowered the threshold for cell death and induced lethal autophagy and apoptosis by inhibiting the ERK pathway.

**Abstract:**

Resistance to cancer therapies remains a clinical challenge and an unsolved problem. In a previous study, we characterized a new colon cancer cell line, namely HT500, derived from human HT29 cells and resistant to clinically relevant levels of ionizing radiation (IR). Here, we explored the effects of two natural flavonoids, quercetin (Q) and fisetin (F), well-known senolytic agents that inhibit genotoxic stress by selectively removing senescent cells. We hypothesized that the biochemical mechanisms responsible for the radiosensitising effects of these natural senolytics could intercept multiple biochemical pathways of signal transduction correlated to cell death resistance. Radioresistant HT500 cells modulate autophagic flux differently than HT29 cells and secrete pro-inflammatory cytokines (IL-8), commonly associated with senescence-related secretory phenotypes (SASP). Q and F inhibit PI_3_K/AKT and ERK pathways, which promote p16^INK4^ stability and resistance to apoptosis, but they also activate AMPK and ULK kinases in response to autophagic stress at an early stage. In summary, the combination of natural senolytics and IR activates two forms of cell death: apoptosis correlated to the inhibition of ERKs and lethal autophagy dependent on AMPK kinase. Our study confirms that senescence and autophagy partially overlap, share common modulatory pathways, and reveal how senolytic flavonoids can play an important role in these processes.

## 1. Introduction

Several types of cancer are associated with aging, including lung, colorectal, prostate, and breast malignancies [1]. Apoptosis and other cell death processes are triggered by the most common cancer therapies, including chemotherapy and radiotherapy [2]. However, cancer is characterized by deregulated apoptotic pathways, hyperactivated survival signaling, accelerated DNA repair mechanisms, and amplified stress response mechanisms. These features, defined as ‘hallmarks of cancer’, promote uncontrolled cell proliferation that results in tumor survival, therapeutic resistance, and cancer recurrence [3,4]. The complexity of cancer cannot be solved by evoking a single mechanism, and the evolution of ‘precision medicine’ in oncology aims at bypassing the resistance to therapy and optimizing cancer treatments based not only on the genetic alterations but also on patients’ biological age that in the field of gerontology and anti-aging medicine, refers to epigenetic alteration and DNA methylation (epigenetic clocks) not always correlated to chronological age [5].

According to recent evidence, autophagy and cell senescence play a role in tumor progression, metastasis formation, and resistance to therapy in vivo. There is increasing evidence that therapy-induced senescence (TIS) and therapy-induced autophagy (TIA) play important and context-dependent roles in cancer survival. However, even if these processes are initially tumor suppressive, they may later influence the tumor microenvironment and contribute to cancer repopulation and metastasis [6,7].

As a result of its central role in preventing cellular damage, autophagy occurs as a natural consequence during and after cancer therapy. Cytoplasmic constituents and organelles (generally defined cargo) are degraded in the lysosome to maintain cellular homeostasis through the synthesis of metabolites, such as glucose from glucagon and amino acids from protein degradation, while the breakdown of ribosomal RNAs, provides a source of nucleotides [8]. Several genes and proteins belonging to the ATG family are essential for the biogenesis of autophagosomes, including those with ubiquitin-like or kinase activities [9]. There are three fundamental metabolic pathways regulated by ATG proteins: amino acid sensing by mTOR kinase complex (mammalian Target of Rapamycin) [10], energy sensing by AMPK (AMP-activated kinase) [11], and stress signaling by HIF (hypoxia-inducing factor) [12]. The TIA is often the primary response of cancer cells to chemotherapy and radiation therapy and has been extensively studied in preclinical and clinical settings. Several stress factors can cause TIA activation and can be either direct or indirect outcomes of cancer treatment, including changes in ROS (reactive oxygen species) concentrations, ATP/AMP ratios, and hypoxia [13]. The onset of senescence or TIS is a typical consequence of genotoxic stress caused by chemotherapeutics such as etoposide or doxorubicin or by radiotherapy. It was initially believed that cellular senescence acted only as an autonomous tumor suppressor [14]. However, senescence involves complex and pleiotropic functions throughout the life cycle, including embryogenesis, cellular reprogramming, tissue regeneration, wound healing, and immunosurveillance [15].

Recent studies have discussed the role of chronic inflammation and tissue microenvironment in cancer therapy [6,16]. The hallmarks of senescent cells include irreversible growth arrest; the expression of a cytoplasmic marker called Senescence Associated β-Galactosidase (SA-βGal), which partially reflects the increase in lysosomal mass [15]; the expression of specific CKD inhibitors (CDKi), namely p16^INK4^ and/or p21^CIP1^, classified *bona fide* as tumor suppressor genes and a robust secretion of numerous growth factors, cytokines, proteases and other proteins (Senescence-associated secretory phenotype, SASP). Specifically, senescence represents an important negative player in cancer therapy, maintaining resistance to apoptosis and supporting chronic inflammation in premalignant tumors [17]. Unfortunately, senescence signaling pathways are still largely unknown.

Natural compounds have been largely explored in recent years for their potential clinical applications with controversial results [18]. Positive examples are the discovery of specific mTOR inhibitors (rapamycin and rapalogs), classically associated with autophagy [19,20,21] and the characterization of “senolytic drugs” for the selective elimination of senescent cells resistant to chemotherapy or radiation therapy. Several promising senolytic agents belong to the class of flavonoids [22,23,24], which are natural bioactive phenolic compounds widely distributed in nature and present in various foods and beverages derived from plants (http://phenol-explorer.eu/compounds/classification; accessed on 3 April 2023). Due to their multiple physiological functions in plant tissues in regulating enzymes involved in cell metabolism and defence mechanisms against foreign agents (radiations, viruses, and parasites), flavonoids have been associated with pleiotropic effects in animal cells. They include more than 4000 different molecules, among them quercetin (Q; 3,3′,4′,5,7-pentahydroxyflavone) [25] and fisetin (F; 3,3′,4′,7-tetrahydroxyflavone) [26] are intensively studied for their senolytic and pro-autophagic activities in different in vitro and in vivo models.

Flavonoids have been described as promising clinical agents due to their relatively low toxicity in vivo, as well as their pleiotropic effect on various biochemical pathways involved in cancer resistance, including apoptosis (CK2, PI_3_K/AKT and ERK signaling), autophagy (AMPK kinase activation), and senescence (p16^INK4^, p21^CIP1^, p27^KIP1^) [25]. Additionally, their synergistic effects in combination with chemotherapeutic drugs or radiation treatment may reduce the systematic toxicity of these treatments [27].

In Burkitt’s lymphoma cells, Q induces autophagic cell death by inhibiting PI_3_K/AKT/mTOR pathways and partially degrading mutant c-Myc [28], as well as inducing lethal autophagy in chronic lymphocytic leukemia cells through an AKT inhibitor called STL-1 [28,29]. In a recent study, Q increased the activity of AMPK kinase in human senescent fibroblasts, resulting in non-apoptotic cell death, a reduction in stress-induced senescent cells (senolytic action), and a suppression of pro-inflammatory responses associated with senescence (low levels of secreted IL-8 and IFN-β, senostatic effects) [30].

As a result of treatment with F and 5-fluorouracil, PI_3_K expression was decreased, AKT and mTOR phosphorylation and their target proteins were reduced, and AMPKα phosphorylation increased in PI_3_K human colon cancer mutant cells [31]. PI_3_K/Akt/mTOR pathway activation was observed in human epidermal keratinocytes treated with F to simulate psoriasis-like disease, inducing differentiation and inhibiting interleukin-22-induced proliferation, as well as activating the PI_3_K/Akt/mTOR pathway [32].

In addition to the promising effects in these preclinical models, both Q and F were recently studied in human clinical trials (phase 1–2) for their senotherapeutic and senolytic effects against different aging-associated diseases such as chronic renal failure, pulmonary fibrosis and osteoarthritis [22,23,24,26,33,34,35,36].

In a previous study, we showed that F and Q in association with IR reduced the expression of p16^INK4^, p21^CIP1^, and synergistically increased cell death compared to IR single treatment in two radioresistant cell lines derived from human osteosarcoma SAOS and colon adenocarcinoma HT-29 [37]. Here, we investigated how Q and F can modulate TIS and TIA in the newly radioresistant HT500 cell line.

## 2. Materials and Methods

### 2.1. Cell Culture and Reagents

The HT29 cell line was used as a model for colorectal cancer that exhibited apoptosis resistance due to TP53 mutation [38]. It was purchased from The American Tissue Culture Collection (ATCC), LGC Standards (Sesto San Giovanni, Milan, Italy) and cultured at 37 °C in a humidified atmosphere with 5% CO_2_ using Dulbecco’s Modified Eagle’s Medium (DMEM) supplemented with 10% foetal bovine serum (FBS; Thermo-Fisher Scientific/Life Technologies, Monza, Italy) with 100 μg/mL penicillin/streptomycin, 2 mM L-glutamine, and 100 μM non-essential amino acids (Thermo-Fisher Scientific/Life Technologies). To avoid any variations caused by the long-term culture of the cells, we used early passages (≤10) to ensure the reproducibility of the results.

We irradiated HT29 cells (1 × 10^6^) with two 5 Gy cycles (the first at time 0 and the second after 24 h) using a GammaCell Elite 1000 instrument (MDS Nordion, Ottawa, ON, Canada) emitting γ-rays and equipped with the ^137^Cesium radioactive source (emitting approximately 2.5 Gy per minute). These doses mimicked the fractionated radiations used in clinical settings (2 Gy in 5 cycles) [37]. In the following three weeks, HT29 cells underwent extensive cell death (>60–80%) measured by using Trypan blue exclusion dye (Merck/Sigma, Milan, Italy). From week 4, some surviving colonies emerged, restored their growth and reached standard levels of cell viability (>90%). We named HT500 the sub-populations derived from parental HT29 cell lines [35]. The surviving cells were then amplified for additional 4 weeks. Afterward, the cells were treated with a trypsin/EDTA solution (Euroclone, Pero, Italy) and counted using Trypan blue exclusion dye (Merck/Sigma) in the automatic cell counter (EveTM, NanoEnTek distributed by VWR, Milan, Italy) in accordance with the manufacturer’s instructions.

To assess the efficacy of senolytic drugs, the cells were pre-irradiated (10 Gy), treated with 40 μM Q or F (Merck/Sigma) or their combination. Crystal Violet dye was used to determine cell viability [39]. In the case of treatment with chloroquine (CLO), it was added to the medium before the above-described protocol to distinguish the effects on autophagy of single and combined treatments.

The selective pharmacological inhibitors of the different pathways investigated were: Sorafenib (Merck/Sigma) for ERK/MAPKs; CAL-101, Idelalisib (Selleck Chem distributed by Fisher Scientific) for PI_3_K/AKT pathway; Rapamycin and Chloroquine (Enzo Life Sciences, Milan, Italy) for mTOR and autophagy flux inhibitors. AMPK kinase allosteric activator, BI-9774, was kindly provided by Boheringer Ingelheim via its open innovation platform opnMe, available at https://opnme.com (accessed on 3 April 2023). Their effects on cell viability were measured using the Cy-Quant assay (Thermo-Fisher Scientific/Life Technologies) [37].

### 2.2. Evaluation of Senescence

Senescence was assessed using two different methods. The first was a quantitative assay kit based on the βGal substrate 4-methylumbelliferyl β-D-galactopyranoside (4-MUG) used to fluorometrically detect SA-βGal enzymatic activity (Enzo Life Sciences). As a result of binding to βGal, the compound 4-MU was produced through the hydrolysis of the substrate and its fluorescence was detected at the wavelengths of 360 nm (excitation) and 465 nm (emission). The protein concentration in cellular lysates was used to normalise the fluorescence of 4-MU for the quantitative determination of SA-βGal activity.

In parallel, classical colorimetric lysosomal (pH 6.0) SA-βGal staining was performed [40]. Briefly, the cells were seeded in 35 mm tissue culture plates at a sub-confluent density (1–1.5 × 10^4^/well). After IR treatment, the cells were treated with 40 μM Q, F or their combination for 72 h, washed with PBS (phosphate-buffered saline solution), and fixed in 3% formaldehyde/PBS (Merck/Sigma) at room temperature for 5 min. Subsequently, the cells were washed with PBS and stained for a period of 30 min in a fresh X-Gal staining solution containing 1 mg/mL of 5-bromo-chloro-indolyl β-D-galactoside (X-Gal; Merck/Sigma) in a buffer consisting of 40 mM sodium phosphate, 150 mM NaCl, 5 mM C_6_FeK_4_N_6_, 2 mM MgCl_2_, pH 6. The cells were incubated for 16 h at 37 °C and subsequently washed with PBS. Microphotographs were taken using a Zeiss-200 invertoscope (Zeiss Axiovert, Zeiss, Jena, Germany; 200× magnification) to determine the positivity to βGal staining.

### 2.3. Analysis of Cytokine Production

Cytokine release (IL-8) was measured by using an in-house sandwich ELISA on the supernatants collected from HT-500 cultures at the end of 72 h incubation. In brief, an aliquot (100 μL) of captured antibody solution (BioLegend, Campoverde S.r.l., Milan, Italy) was plated into ELISA wells (Nunc Maxisorb; eBioscience Inc., San Diego, CA, USA) and incubated overnight at 4 °C. After the removal of the antibody solution, 200 μL of PBS supplemented with 1% BSA (Bovine Serum Albumin, blocking buffer) was added to each well and incubated at room temperature for 2 h. Next, cytokine standard and samples diluted in blocking buffer supplemented with 0.05% Tween-20 were added to the respective wells and incubated overnight at 4 °C. At the end of the incubation, 100 μL of biotinylated antibody solution (BioLegend) was added to the wells and incubated for 2 h at room temperature. Streptavidin–horseradish peroxidase conjugate (1:1500 dilution; BioLegend) was then added to the wells and incubated for 1 h at room temperature. Finally, 200 μL of 63 mM Na_2_HPO_4_, 29 mM citric acid (pH 6.0) containing 0.66 mg mL^−1^ o-phenylenediamine/HCl and 0.05% hydrogen peroxide were dispensed into each well, and the wells were allowed to develop. The absorbance was read at 450 nm by using an iMark micro-plate reader (Bio-Rad Laboratories), and the cytokine concentrations, calculated by the use of proper standard curves, were finally expressed as pg mL^−1^.

### 2.4. Measurement of Autophagy

HT500 cells were irradiated (10 Gy) and incubated for 96 and 120 h with Q, F (40 μM) or their association with IR as described in the figure legends. The compounds were freshly added when the cell medium was replaced (48 h). Autophagy activation was detected by using the Cyto-ID autophagy detection kit based on the use of a cationic amphiphilic tracer able to specifically detect the number of intracellular autophagosomes (Enzo Life Science) [41]. The 488 nm-excitable green dye has been optimized for minimal staining of lysosomes while exhibiting bright fluorescence upon incorporation into pre-autophagosomes, autophagosomes, and autolysosomes. After incubation, the cells were washed, and the mixture containing the autophagy detection marker (Cyto-ID) and nuclear dye (Hoechst 33342) was added. The cells were washed with assay buffer before being photographed using a fluorescence microscope (Zeiss Axiovert 200; 400× magnification). The autophagosomes were quantified by normalizing green fluorescence (Cyto-ID) and blue fluorescence (Hoechst) using a microplate fluorescence reader (Synergy HT BioTek, Milan, Italy).

### 2.5. Measurement of Apoptosis

To determine caspase-3 enzymatic activity, HT500 cells were pre-irradiated (10 Gy) and then treated, respectively, for 72 h with 40 μM Q or F or their combination with IR. Briefly, after treatments, the cells were collected and centrifuged at 400× *g* for 5 min, washed with PBS and lysed in lysis buffer (10 mM Hepes, pH 7.4; 2 mM ethylenediaminetetraacetic acid; 0.1% [3-(3-cholamidopropyl) dimethylammonio]-1-propanesulfonate, 5 mM dithiothreitol, 1 mM phenylmethylsulfonylfluoride, 10 μg/mL pepstatin-A, 10 μg/mL apronitin, 20 μg/mL leupeptin). The reaction buffer and the conjugated amino-4-trifluoromethyl coumarin substrate (AFC): benzyloxycarbonyl-Asp (OMe)-Glu (OMe)-Val-Asp (OMe)-AFC (Z-DEVD-AFC) (Enzo Life Science) were added to the cellular extracts (10 μg) and incubated at 37 °C for 30 min. Caspases-3 proteolytically cleaved the substrate, and the free fluorochrome AFC was detected through the use of a multi-plate reader (Synergy HT BioTek) with an excitation of 400 ± 20 nm and emission at 530 ± 20 nm. An AFC standard curve was used in order to quantify the activity of the enzyme. Caspase-3 specific activity was expressed as nmol of AFC produced per min per μg proteins at 37 °C in the presence of saturating concentrations of the substrate (50 μM). The determination of apoptotic nuclei was performed by counting in each sample, a minimum of 100–200 cells in duplicate in order to determine the percentage of apoptotic bodies, as previously described [28].

### 2.6. Immunoblotting

Using the Bradford method, the protein concentration of the cells was determined after they were lysed in a buffer containing protease inhibitors as well as phosphatase inhibitors to inhibit proteases and phosphatases [42]. The total lysates (20 μg/lane) were loaded on a 4–12% precast gel (Novex Bis-Tris precast gel; Thermo-Fisher Scientific/Life Technologies) using 50 mM MES (2-(Nmorpholino) ethanesulfonic acid) buffer at pH 7. In some experiments, protein lysates (20 μg) were added with 2× Laemmli loading buffer, heated at 95 °C, were incubated for about 16 h with the following primary antibodies diluted 1:1000 in 3% BSA/T-TBS: anti pAKT (cat. # 4058,), anti pERK1-2 (cat. # 4370S), anti-pAMPKα^Thr172^ (cat. # 2535S), anti-AMPKα (cat. # 5832S), anti pULK1/ULK1 (cat. # 5869/8054), anti LC3I/II (cat. # 12741S) and p62^SQST^ (cat. # 5114S) from Cell Signaling Technologies, anti AKT (cat. # GTX121937) from Genetex (Prodotti Gianni, Milan, Italy), anti ERK1/2 (cat. # SC-093) was from Santa Cruz Biotechnologies (SC-093); anti α-Tubulin was from MERK-Millipore (cat. # T9026). Following washing with T-TBS, the membranes were incubated for 2 h with a secondary antibody linked to horseradish peroxidase (diluted 1:20,000 in T-TBS). The immunoblots were developed using the ECL Prime Western blotting detection system kit (GE Healthcare, Milan, Italy). Band intensities were quantified and expressed as optical density on a Gel Doc 2000 Apparatus (Bio-Rad Laboratories, Milan, Italy) and Multianalyst software (Bio-Rad Laboratories).

### 2.7. Statistical Analysis

One-way ANOVA followed by Turkey’s or Bonferroni’s multiple comparisons test was performed using GraphPad Prism version 9.5.1 for Windows, GraphPad Software, San Diego, CA, USA, www.graphpad.com (accessed on 3 April 2023). Statistical significance was accepted at a *p*-value of less than 0.05. The results have been expressed as mean ± standard deviation (s.d.) based on values obtained from independent experiments performed in duplicate, triplicate or quadruplicate. Specific values were indicated in figure legends as follows: * *p* < 0.05, ** *p* < 0.01, *** *p* < 0.001; **** *p* < 0.0001.

## 3. Results

### 3.1. Comparison of Cellular and Biochemical Markers in HT29 and HT500 Cell Lines

In a previous study, we irradiated HT29 cells (1 × 106) with two 5 Gy cycles (the first at time 0 and the second after 24 h) with sub-lethal doses of IR. These doses mimicked the fractionated radiation doses used in clinical settings (2 Gy in five cycles) [37]. In the following three weeks after irradiation, HT29 cells showed extensive cell death (>60–80%) verified by using the Trypan blue exclusion dye count. From week 4, some surviving colonies restored their growth and reached standard levels of cell viability (>90%). We named HT500 the subpopulations derived from parental HT29 cell lines [35]. To avoid changes in cellular and biochemical parameters due to long permanence in culture, in all experiments, HT500 cells were used in the same early passages (<10).

We showed that HT500 cells were characterized by higher levels of senescence markers p16^INK4^ and p21^CIP1^ compared to parental HT29 and elevated levels of basal SA-βGal as a consequence of TIS, which was associated with increased resistance to oxidative stress in radioresistant cells [37]. These characteristics are summarized in Table 1. Therefore, we selected HT500 cells as a validated model of radioresistance to study the senolytic and autophagic effects of Q and F.

### 3.2. Quercetin and Fisetin Show Senolytic and Anti-SASP Activity in Radioresistant HT500 Cells

HT500 cells were pre-irradiated (10 Gy) and then treated with 40 μM Q, F or their combination for 72 h. The SA-βGAL activity was quantified using a fluorometric kit (Figure 1a) while micrographs (Axiovert 200, 200× magnification) of the same samples (Figure 1b) were obtained with classical SA-βGAL staining [40]. In a previous study, we verified by immunoblotting analysis the expression of p16^INK4^ and p21^CIP1^ senescence markers in HT500 cells after 72 h of treatment with IR, F and Q or their combination [37]. The results obtained with a quantitative SA-βGAL assay confirmed that Q and F in single treatment effectively reduced SA-GAL with respect to IR and untreated cells (40–30%), while Q or F plus IR reduced SA-GAL by about 50%.

We investigated whether IR induced the production of pro-inflammatory cytokines as the result of TIS and whether Q and F could inhibit their secretion in cell culture medium by showing anti-SASP effects. To determine IL-8 levels, we measured the effects of different treatments through the use of a sandwich ELISA at the same time-points (72 h), as indicated in Figure 1a. In fact, Figure 1c confirmed the increase in IL-8 production (>55%) after IR as a result of the increase in SA-βGAL. Q and F alone were able to reduce IL-8 (<30–40%, respectively), and the decrease was significantly lower in the combined treatment, IR plus Q and IR plus F, with respect to IR (<60–70%) mirroring the senolytic effect observed with the SA-βGAL quantitative assay.

### 3.3. IR Differently Modulates the Autophagic Flux in HT29 and HT500 Cells

Radiotherapy response in colon cancer is not solely correlated with senescence. We hypothesized that TIA might contribute to radioresistance [7]. Therefore, we evaluated the expression of autophagic markers, e.g., LC3-I/II and p62^SQSTM1^ [43], in parental HT29 and radioresistant HT500 cells. In Figure 2, both HT29 and HT500 cells showed high LC3-II expression at basal levels after IR for 24 to 72 h. HT29 cells expressed significantly lower p62^SQSTM^ levels than HT500 cells after 24 h, and the opposite occurred for LC3-II expression as well. This pattern was reasonably due to the different modulation of the autophagy flux in HT500 compared to parental HT29 cells, as we recently demonstrated in SAOS osteosarcoma cells [44]. We also observed that senescence markers decreased after 72 h of IR, suggesting that autophagy was the predominant phenotype present in radioresistant cells over time. For these reasons, we decided to study how the natural flavonoids Q and F could modulate this process.

### 3.4. Quercetin and Fisetin Modulate Autophagic Flux in HT500 Cells

To confirm the capacity of Q and F to modulate the autophagic flux in HT500 cells, we measured several autophagic markers after treatment with IR, F, Q, and their combinations in HT500 cells with the quantitative fluorometric Cyto-ID assay. Autophagosome stained with Cyto-ID green dye and nuclei stained with Hoechst (blue) (Figure 3) show that IR was able to increase (>20%) autophagosome associated fluorescence (green dots in panel c) expressed as FITC/DAPI ratio with respect to untreated cells after 96 h. A similar increase in autophagosome-associated fluorescence was measured after treatment with Q and F (>8% and >15%, respectively). However, when IR was associated with Q and F, the cytoplasmic autophagosome increased significantly (>22% and >25%, respectively) with respect to single treatments. The same trend was observed after 120 h, even if the baseline control was higher, according to the dynamic of the autophagic flux.

### 3.5. Quercetin and Fisetin Enhance Type I (Apoptosis) and Type II (Autophagic) Cell Death in HT500 Cells

To clarify the nature of autophagy, e.g., protective vs. cytotoxic [45], chloroquine (CLO; 10 μM) was added to the cell culture medium as a pharmacological inhibitor of autophagy, and cell viability was evaluated in HT500 cells after treatment with IR, Q, F, and their combinations at 96 h (Figure 4a).

A protective form of autophagy was induced in HT500 cells by IR in the presence of CLO, resulting in a slight increase in cell death (>14%). When treatments with Q or F alone were compared with the combinations CLO plus Q or CLO plus F, a clear protective role of autophagy was observed (>50% for Q; >45% for F) (Figure 4a). It is notable, however, that the cytotoxic effects of IR plus Q and IR plus F were significantly reduced when CLO was present in the medium (<35% and <39%, respectively), indicating that autophagy, under these conditions, exerted a cytotoxic effect. Frequently, lethal autophagy (type II cell death) is associated with apoptosis (type I cell death). This phenomenon was confirmed by measuring caspase-3 enzymatic activity (Figure 4b), which slightly increased in HT500 cells 72 h after IR, confirming that apoptosis is not the main consequence associated with the IR response. However, caspase-3 activity was significantly higher (>20–30%) in the combined treatments IR plus F and IR plus Q if compared to IR single treatment. The analysis of apoptotic bodies confirmed this result. In fact, the presence of nuclear fragmentation (a marker of apoptotic cell death) after IR and the detection of apoptotic bodies were observed only in 14% of irradiated cells after 120 h, while in the combined treatments, IR plus F or IR plus Q, they were significantly higher (>1.5–2-fold) with respect to irradiated cells (Figure 4c,d).

These data suggest that only the association between γ-rays and natural senolytic agents reached the threshold necessary to induce cell death (>about 60% after 96 h, Figure 4a) through the selective elimination of senescent cells (senolysis).

### 3.6. Quercetin and Fisetin Target Different Signaling Pathways Associated to the Senolytic Effects and Autophagy Induction in HT500 Cell Line

We previously demonstrated that Q could inhibit the PI_3_K/AKT pathway [28,46], whose hyper-activation is often associated with proliferation and/or resistance to apoptosis in cancer cells [47]. Therefore, we performed an immunoblot to detect the phosphorylated and active form of AKT (pAKT^Ser472^) in HT500 cells shortly after (20 min) the treatment with Q and F (40 μM). Figure 5a,c) shows that both Q and F can downregulate phosphorylation of AKT in Ser^472^ (<about 40%) with respect to untreated cells.

Subsequently, we examined the contribution of ERK1-2/MAPK and AMPK pathways to radioresistance in the HT500 cell line in terms of senescence and autophagy modulation.

The ERK1-2/MAPK and AMPK signaling pathways were selected because they represent well-known intracellular targets of flavonoids in cancer cells triggering apoptotic resistance, autophagy and senescence [11,20,30,48,49,50]. Taking advantage of the rapid intracellular uptake of Q [46], we measured changes in the activation of ERK1/2 and AMPK at early time points (20 min) in HT500 cells. Figure 5b,d shows that both Q and F were able to downregulate the phosphorylation levels of ERK1/2 (pERK1/2^Thr202^/^Tyr204^) by about 50% for both Q and F. In parallel, according to the role of F and Q in inducing autophagy in HT500 cells (Figure 4), we detected a strong up-regulation of AMPK kinase revealed through the increased expression of its phosphorylated form on Thr^172^ embedded in the catalytic subunit α/α1 of AMPK (Figure 5e,g). Densitometric analysis showed that the expression of pAMPKα^Thr172^ was about 7- and 6-fold higher for Q and F, respectively, compared to untreated controls after 20 min of treatment. These values were comparable to the AMPK allosteric activator BI-9774.

We also measured the phosphorylation status of ULK1 in Ser555, it being a part of the ATG1 kinase complex, which represents the most upstream component of the core autophagy machinery conserved from yeast to mammals and a direct substrate of AMPK kinase [51]. The data in Figure 5f–h confirm that both flavonoids enhanced ULK1 kinase activity (>4–6 times higher with respect to untreated cells) and, consequently, activated the autophagy machinery.

### 3.7. Quercetin and Fisetin in Association with γ-Ray Downregulate ERK/MAPK and Activate AMPK Kinases

To evaluate whether Q and F were able to down-regulate ERK/MAPKs and activate AMPK soon after IR, we performed an immunoblot analysis incubating HT500 cells with the single flavonoids and their combination with γ-rays. Figure 6a,b) confirms that Q and F were able to significantly down-regulate ERK/MAPK activation after IR (20 min). We also showed that even if IR activated AMPK kinase (>2-fold with respect to untreated cells), Q and F were more efficient in increasing AMPK activity, both in single treatments or in combination with IR (>7 and 5-fold with respect to untreated cells) (Figure 6c,d).

### 3.8. Pharmacological Modulation of mTOR and AMPK Kinases Confirm the Role of Autophagy in Sensitizing Radioresistant HT500 Cells

Autophagy induction in cell death-resistant cancer cells can be a protective phenomenon [20]. We reasoned that pharmacological activators of autophagy, e.g., Rapamycin for the mTOR pathway and BI-9774 for AMPK, could complement Q and F in bypassing radioresistance in HT500 cells. As shown in Figure 7, we performed a cell viability assay using Cy-Quant fluorescent dye, and we demonstrated that both autophagy inducers can bypass cell death resistance in HT500 cells, resulting in a 60% reduction in cell viability when IR plus Rapamycin or IR plus BI-9774 were combined. Interestingly, HT500 cells presented apoptotic bodies when IR was combined with Rapamycin (Figure 7b).

### 3.9. Pharmacological Inhibition of ERK/MAPKs or PI_3_K/AKT Signalling Pathways Induces Cell Death in Radioresistant Cells

The role of ERK/MAPKs and PI_3_K/AKT signaling pathways in cell death resistance to IR was validated by treating HT500 cells with well-known pharmacological inhibitors of these pathways. To bypass resistance to cell death in colon cancer cells, we used Sorafenib, an ERK/MAPK inhibitor approved for renal cell and hepatocellular carcinomas, in combination with Metformin, an AMPK activator [52]. We also used CAL-101, a drug targeting the p110δ subunit of PI_3_K, which demonstrated clinical efficacy against chronic lymphocytic leukemia and colon cancer cells regardless of p53 mutational status [53]. In HT500 cells, Sorafenib (20 μM) and CAL-101 (10 μM) inhibition of ERK/MAPK and PI_3_K signaling pathways, respectively, was successfully observed through immunoblotting (Figure 8a,b). After irradiation (10 Gy) and incubation with Sorafenib or CAL-101 or their combined treatments with IR for 72 h, cell viability was assessed using the Cy-Quant assay. Even though both drugs can reduce cell viability in HT500 cells (Figure 8c), only the combination of IR plus Sorafenib or CAL-101 can effectively induce cell death in radioresistant cells of about 50–60%. In addition, apoptotic bodies were detectable after IR plus Sorafenib and IR + CAL-101 treatments (Figure 8d, white arrowheads).

## 4. Discussion

Studies based on natural compounds, such as flavonoids, demonstrated that the hallmarks of cancer, such as apoptosis, autophagy, and senescence, could intercept the hallmarks of aging [54]. The novelty of the present paper resides in the observation that both senescence and autophagy can be induced by IR treatment, possibly due to common modulators shared by these two processes.

The complexity and context-dependent nature of autophagy in cancer led to the creation of a new field called ‘oncophagy’, which aims to identify efficient inducers of autophagy that are expected to act at the right time and place during cancerogenesis or cancer therapy [20]. However, the identification of these agents is still in its infancy, even if natural compounds, such as rapalogs and some polyphenols (e.g., resveratrol and curcumin), represent good candidates.

In the present study, we demonstrated that autophagic flux was modulated differently by IR in HT29 cells compared to their radioresistant derivative HT500. We observed that basal autophagic levels were comparable in both models, but p62^SQTSM^ expression was always higher in HT500 cells with respect to parental HT29 cells (Figure 2). It is possible that the simultaneous presence of protective phenotypes (TIS and protective autophagy) is responsible for repairing cellular damage due to IR, as demonstrated in other cases of radioresistant cells [44] and summarized in Table 1.

According to a recent study, the homeostatic state of senescence is coordinated through selective autophagy of specific regulatory components. The p62^SQSTM1^-dependent elimination of Keap1 promotes Nrf2 translocation and its transcription of antioxidant genes, maintaining redox homeostasis during senescence [55]. Interestingly, HT500 cells show higher levels of p62^SQSTM1^ after 24 h of IR compared to HT29 cells (Figure 2). In human osteoarthritis, a model of pathology associated with senescence, selective autophagic networks are clearly observed in vivo, which is consistent with our results [56].

Figure 3 shows that TIA is still present in the HT500 cell line after 96–120 h of IR when senescence begins to decline. We verified in HT500 cells that not only IR but also Q and F induce a protective form of autophagy (Figure 4). However, in the presence of the combined treatment, the level of autophagy is significantly higher (Figure 3) and an “autophagic switch” occurs, causing the transition from a protective form of autophagy to the not protective or lethal one, as described in different cellular models [28,57,58]. Both TIS and TIA may explain HT500 radioresistance through long-term regulation of intracellular redox status (ROS and GSH). Further experiments will clarify the biochemical pathway, possibly involving the p62^SQSTM1^/Nrf2/Keap1 system, which is responsible for the protective role of senescence and autophagy against the ROS-mediated cytotoxicity induced by IR [55].

Recent studies evidenced that the effectiveness of senolytics is strictly dependent on the time of administration [56]. In fact, as summarized in Table 1 and in Appendix A, we observed that HT500 cells were more sensitive to combined treatment of Q plus IR compared to the parental cells when the senolytics were added 72 h after irradiation. Interestingly, in HT500 cells, clear cytotoxic effects were achieved when F and Q were added immediately after irradiation, consisting of apoptosis (type I cell death) and cytotoxic autophagy (type II cell death) (Figure 4).

Confirming the selectivity of senolytic agents, the radiosensitizing effect of F was absent in the same experimental condition in parental HT29 cells (Table 1 and Appendix A). These results also confirm previous studies in which F did not induce apoptosis in p53-null HCT116 and p53-mutant HT-29 colon cancer cells. Similarly, the apoptotic effects of F in a single treatment or in combination with IR were clearly observed in xenograft mice implanted with murine p53-wild type CT26 colon cancer cells [59,60,61]. Furthermore, in these studies, very high concentrations of F (>100 μM) were applied to induce cell death. From a clinical point of view, these concentrations are very difficult to reach due to the well-known low bioavailability and metabolic transformation of flavonoids in vivo [25,62]. Our results validate the hypothesis that natural senolytics, being unstable in cell culture medium and poorly bioavailability, are most effective when used in a hit-and-run modality, which, at least in vivo, may represent a procedure that avoids potential toxicity and off-target effects [56].

More intriguing is the senolytic molecular mechanism(s) of action of F and Q in the radioresistant cells here investigated. These compounds can directly or indirectly modulate the activity of multiple kinases associated with SASP, senescence, or autophagy, such as the PI_3_K/mTor/AMPK or ERK/MAPK pathways after γ-ray irradiation [6,32,63,64]. Q and F act soon after incubation (20 min) on the PI_3_K/AKT pathways, revealing that HT500 cells rely on this proliferative signaling (Figure 5). Previous studies demonstrated that Q is a direct inhibitor of PI_3_K and CK2 kinases in a chronic lymphocytic leukemia model [46]. Therefore, the present data suggest that these kinases may also be the primary and direct targets of Q in HT500 cells. However, 60 min after IR or after the incubation with Q, the inhibitory effect disappeared (Appendix A). This unexpected result deserves future investigations, but it can be due to the activation of redundant survival pathways.

Our results also showed that in HT500 radioresistant cells, both flavonoids, Q and F, alone or soon after IR clearly downregulated the ERK/MAPKs pathway, often associated with proliferation, inflammation, and apoptotic resistance [65]. Different studies have demonstrated that IL-8 and its receptor CXCR2 are two of the most significantly upregulated chemokines in colorectal cancer. IL-8, by binding to its receptors, can act not only on inflammatory responses and infectious diseases but also on cancer cells through their receptors to promote migration, invasion, proliferation, and in vivo angiogenesis [66]. Therefore, IL-8 and CXCR2 may be important therapeutic targets against cancer. Kim et al. demonstrated that the IL-8 levels of mRNA and protein were significantly increased in tamoxifen-resistant breast cancer cells compared to sensitive cells. Elevated expression of IL-8 mRNA in resistant cells was suppressed by a specific MEK1/2 inhibitor, UO126, but not by the specific PI_3_K inhibitor LY294002. On the contrary, the over-expression of constitutively active MEK significantly increased the levels of IL-8 mRNA expression in cancer-sensitive cells [67]. Our results confirm that, in an advanced-stage colon cancer model, Q and F can bypass cell death resistance by interfering with the MEK/ERK/IL-8 signaling (Figure 1; Figure 6).

Although hyperactive MAPK signaling has a dominant role in cancer biology, it is fine-tuned by other signaling, such as PI_3_K/AKT/mTOR and AMPK, during disease progression [65]. Recent studies investigated the complicated interplay between MAPK and AMPK signaling in cellular carcinogenesis and their implications in cancer therapies, revealing that AMPK signaling can reversibly regulate hyperactive MAPK signaling in cancer cells by phosphorylating its key components, RAF/KSR (kinase suppressor of Ras) family kinases [65]. In contrast, ERK and ribosomal protein S6 kinase A (RSK), two downstream kinases of MAPK signaling, have been shown to phosphorylate and inhibit the upstream activator of AMPK, LKB1, and thus block the activation of AMPK by LKB1 in BRAF^V600E^-driven melanoma. Our results (Figure 6) confirm the existence of this complex interplay. Considering the emerging roles of the AMPK and mTOR complex on the molecular pathways that connect autophagy and senescence, they may represent ideal targets of natural flavonoids in these processes [6,7]. The concept that inhibiting ERK/MAPKs and/or activating AMPK may provide a promising strategy to reverse the onset of radioresistance in cancer cells [68] is reinforced by our observation (Figure 7a and Figure 8a) that Sorafenib and BI-9774 exert radiosensitising effects by inhibiting ERK/MAPK signaling and activating AMPK, respectively.

Interestingly, rapamycin, a natural inhibitor of the mTOR pathway and CAL-101, a selective inhibitor of PI_3_Kδ isoform, can sensitize HT500 cells to radiation therapy (Figure 7; Figure 8). It cannot be excluded that, according to the literature [63], F or Q may inhibit mTOR directly and indirectly activate AMPK or inhibit kinase(s) upstream PI_3_K/AKT and ERK/MAPKs. Immunoblot analyses revealed that AMPK activation was measurable after only 10 min from Q and F treatment, while ERKs and AKT inhibitions were detectable after 20 min (Figure 5). Based on these observations, we can hypothesize that LKB1 kinase may represent a putative target of Q or F, being an upstream activator of AMPK [69].

Our data are also in agreement with the concept of functional pleiotropy of flavonoids [64], i.e., their ability “to hit two birds with a single stone”. From a pharmacological point of view, “pleiotropy” could be interpreted as synonymous with poor selectivity; however, our study reveals a novel and specific molecular mechanism of natural senolytics that deserves future in vivo investigations.

In summary, radioresistant cells represent a feasible model of TIS and TIA, two processes, as we discussed, functionally heterogeneous and context-dependent (tissue of origin, time after insult, genetic background), but always associated with anti-apoptotic and cell death-resistant phenotypes. On the other side of the coin, TIS may display a therapeutic opportunity because the clearance of senescent cells by senolytics (first hit) can restore the sensitivity to cell death (second hit). From a molecular point of view, we hypothesize that AMPK activation can link cancer cell response to radiation treatment [69] through induction of senescence and, later, autophagy, but the concomitant ERK/MAPKs inhibition by flavonoids can lower the cell death threshold and induce lethal autophagy and apoptotic cell death (Figure 9).

In clinics, the use of F and Q as senolytics in adjuvant radiation therapy may reduce total radiation exposure and improve patient quality of life during and after radiation treatment [33,35]. In fact, in vivo treatment with the senolytic cocktail, Desatinib plus Q mitigates radiation ulcers, which represent a common toxic side effect in patients receiving radiation therapy [70].

More research into the optimal combination therapy for colorectal cancer is desirable, but additional efforts are needed to define the features of radioresistant cellular subpopulations in depth, such as the presence of cancer stem cells and the secretion of other pro-inflammatory molecules frequently associated with SASP (e.g., IL-6, TNF-α). Hopefully, future studies will reveal the exact molecular links between redox intracellular status and biochemical pathways (AMPK/ULK1/mTor) associated with senescence and autophagy [71] and clarify how natural flavonoids can function in vivo as senolytic adjuvants to kill cells that escape radiotherapy-induced cell death.

## 5. Conclusions

The present study confirms that TIS and TIA are often associated with cell death resistance in cancer. In this context, only the association between γ-rays and natural senolytic agents reached the threshold necessary to induce type-I/II cell death throughout the selective elimination of senescent cells (>60%, senolysis) in HT500 cell line, a new model of radioresistant colon cancer cells. This work attempts to clarify how the natural flavonoids, Q and F, can function as senolytic adjuvants capable of killing cancer cells only when associated with radiotherapy. Due to the pleiotropic effects of Q and F, their biochemical targets include pathways associated with resistance to apoptosis and inflammatory status (MEK/ERK/IL-8) but also autophagy and senescence (AMPK/ULK1).

## Figures and Tables

**Figure 1 cancers-15-02660-f001:**
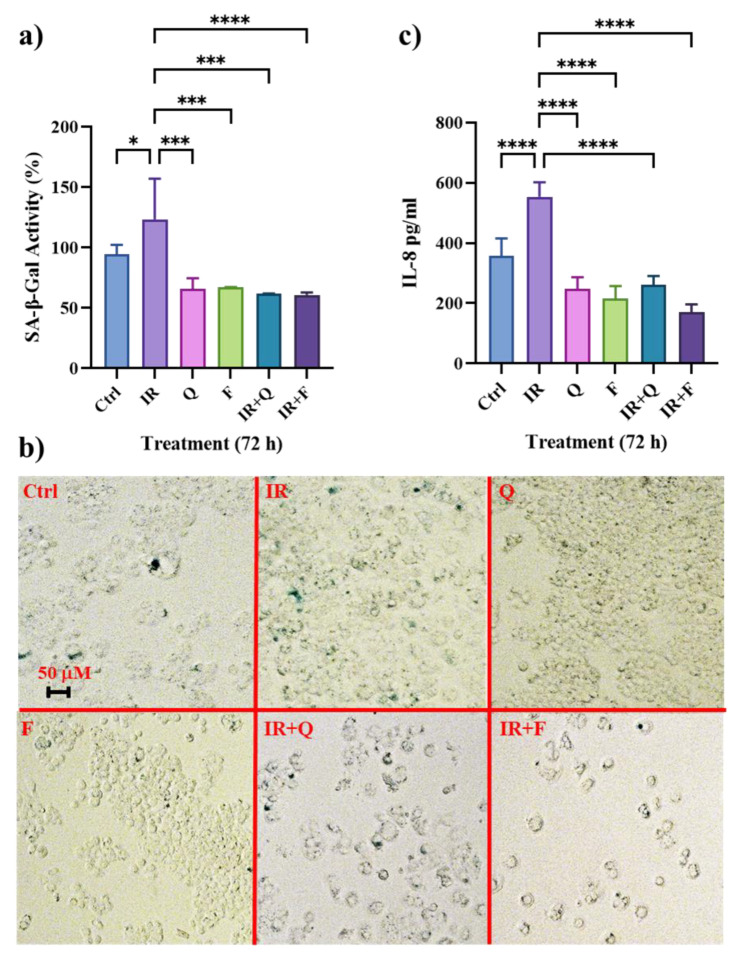
Senescence markers in the HT500 cell line following senolytic treatment. Quantitative measurement of SA-βGAL activity in HT500 cells obtained with a fluorometric kit (panel (**a**)) and micrographs (Axiovert 200; 200× magnification) of the same experimental samples (panel (**b**)) obtained with classical SA-βGAL staining as described in Methods section. Bar graphs represent the mean of three experiments in duplicate ± s.d. Symbols indicate significance: * *p* < 0.05, *** *p* < 0.001, **** *p* < 0.0001 (One-way ANOVA). Panel (**c**): analysis of IL-8 production 72 h after IR, Q, F mono-treatment or their combination. Culture supernatants were collected after 72 h and analyzed by using sandwich-type ELISA. Columns represent the mean ± s.d. and are representative of two independent experiments in duplicate. Symbols indicate significance: **** *p* < 0.0001 (one-way ANOVA).

**Figure 2 cancers-15-02660-f002:**
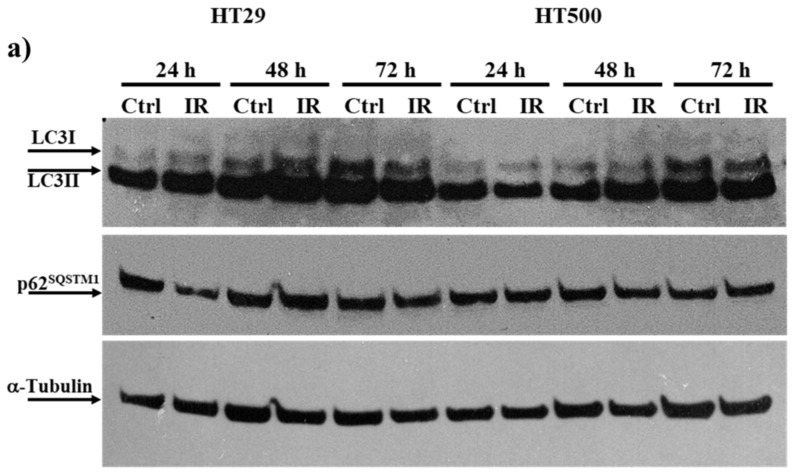
IR modulates autophagic flux in HT29 versus HT500 cells. Panel (**a**): cells were irradiated at the indicated time points with 10 Gy. Cellular proteins were subjected to SDS/PAGE and immunoblot analysis of autophagic markers LC3-I/II and p62^SQSTM^. Panels (**b**,**c**): band intensities were quantified measuring optical density on Gel Doc 2000 and analyzed by Multi-Analyst Software. Values in the histograms indicate protein expression normalized with respect to α-Tubulin (mean of two experiments ± s.d.) Symbols indicate the significance: * *p* < 0.05, ** *p* < 0.01; *** *p* < 0.001 (One-way ANOVA). The uncropped blots and molecular weight markers are available in the Appendix A.

**Figure 3 cancers-15-02660-f003:**
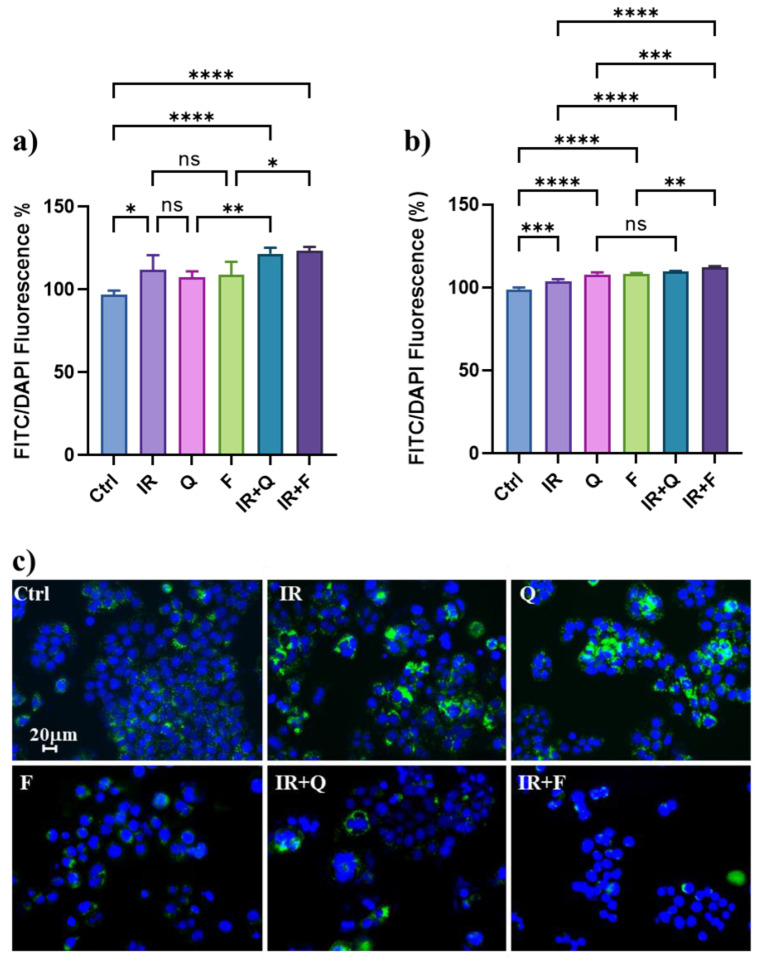
Cells were irradiated (10 Gy) and then treated for 96 h (panel (**a**)) and 120 h (panel (**b**)) with Q (40 μM), F (40 μM) or IR plus the Q and F. The medium was removed, and cells were incubated 30 min with Cyto-ID autophagosome-specific dye and Hoechst nuclear stain as described in the Methods section. The green fluorescence of the cytoplasmic autophagosome was normalised with the blue fluorescence emitted by the nuclei and expressed as a percentage of Ctrl (mean of two experiments ± s.d.). Symbols indicate significant differences: * *p* < 0.05, ** *p* < 0.01; *** *p* < 0.001, **** *p* < 0.0001 (One-way ANOVA with Bonferroni’s multiple comparisons test was used in these experiments). Panel (**c**): micrographs (400× magnification) of the representative field taken in different well plates where HT500 cells were treated as described in (**a**).

**Figure 4 cancers-15-02660-f004:**
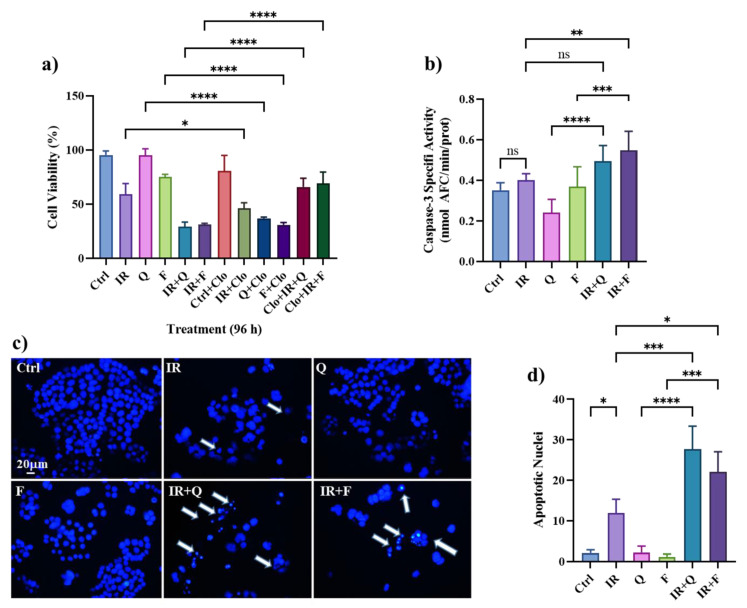
Different modulation of autophagy and apoptosis by IR, Q, and F in HT500 cells. Panel (**a**): Crystal violet viability assay in HT500 cells treated as described with or without 10 μM chloroquine (CLO) to discriminate the autophagic role of single and combined treatments. Symbols indicate significant differences: * *p* < 0.05, ** *p* < 0.01; *** *p* < 0.001, **** *p* < 0.0001 (One-way ANOVA with Bonferroni’s multiple comparisons test was used in these experiments). Panel (**b**): cells were pre-irradiated (10 Gy) and then treated, respectively, for 72 h with 40 μM Q or 40 μM F or their combination with IR, then Caspase-3 enzymatic activity was measured as described in the Methods section. Symbols indicate significant differences (mean of two experiments in duplicate ± s.d.): n.s. not significant, ** *p* < 0.01; *** *p* < 0.001, **** *p* < 0.0001 (One-way ANOVA). Panels (**c**,**d**): cells were pre-irradiated (10 Gy) and then treated, respectively, for 120 h with 40 μM Q or 40 μM F or their combination with IR. Hoechst-stained apoptotic nuclei were photographed (panel (**b**)) and counted (panel (**c**)) using fluorescence microscopy (>100 cells/field, 400× magnification; mean of two experiments in duplicate ± s.d.). White arrows indicate apoptotic nuclei. Symbols indicate significant differences: * *p* < 0.05, ** *p* < 0.01; *** *p* < 0.001, **** *p* < 0.0001 (One-way ANOVA).

**Figure 5 cancers-15-02660-f005:**
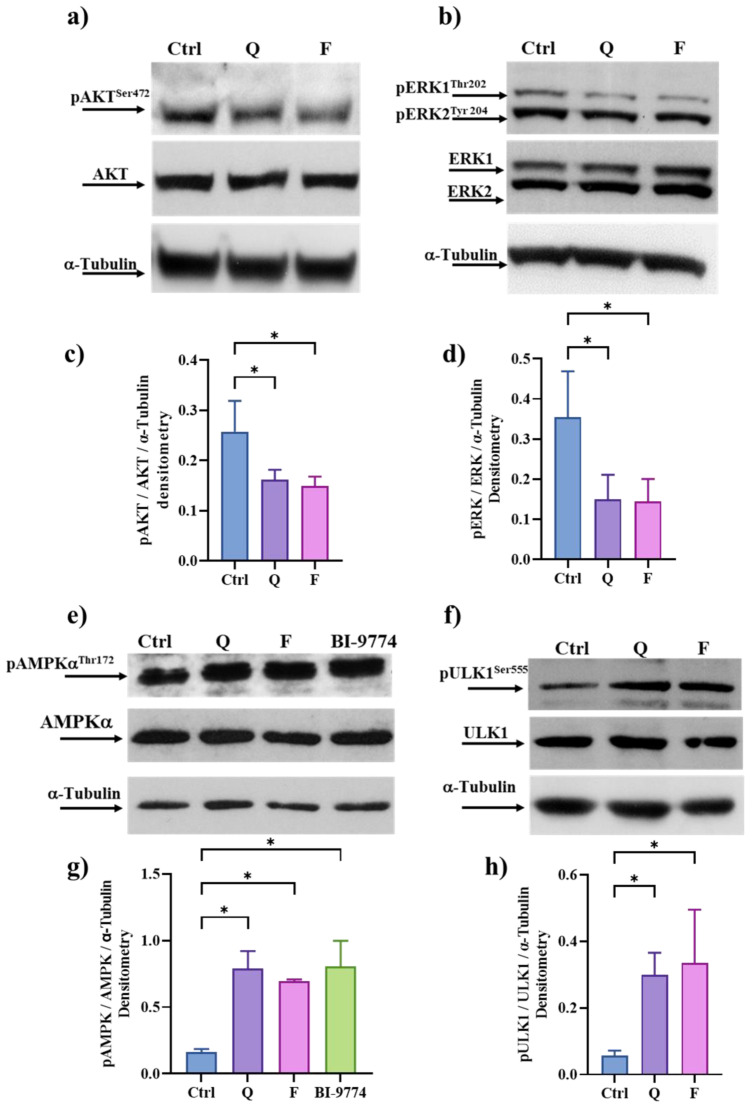
Western blot analysis of p-AKT-AKT (panels (**a**,**c**)), pERK1/2-ERK (panels (**b**,**d**) pAMPK-AMPK (panels (**e**,**g**)) and pULK1-ULK1 proteins (panels (**f**,**h**) in HT500 cells following 20 min of treatment with 40 μM Q or F respect to vehicle (0.2% DMSO). BI-9774 (10 μM) was used as a positive control for AMPK activation. Band intensities were quantified by measuring optical density on Gel Doc 2000 and analyzed by Multi-Analyst Software; values in the graphs indicate protein expression normalized with respect to α-Tubulin (mean of two experiments ± s.d.). Symbols indicate significant differences * *p* < 0.05 (one-way ANOVA). The uncropped blots and molecular weight markers are available in the Appendix A.

**Figure 6 cancers-15-02660-f006:**
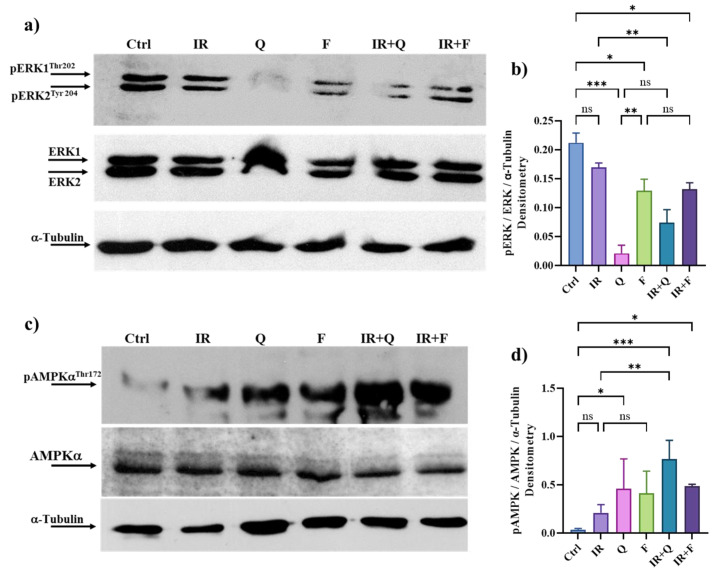
Western blot analysis of pERK1/2-ERK (panels (**a**,**b**)) and pAMPK-AMPK (panels (**c**,**d**)). HT500 cells were irradiated with 10 Gy and then incubated for 20 min with Q, F or the combined treatments. Band intensities were quantified by measuring optical density on Gel Doc 2000 and analyzed by Multi-Analyst Software; values in the graphs indicate protein expression normalized with respect to α-Tubulin (mean of two experiments ± s.d.). Symbols indicate significant differences: * *p* < 0.05, ** *p* < 0.01; *** *p* < 0.001, n.s. not significant (one-way ANOVA). The uncropped blots and molecular weight markers are available in the Appendix A.

**Figure 7 cancers-15-02660-f007:**
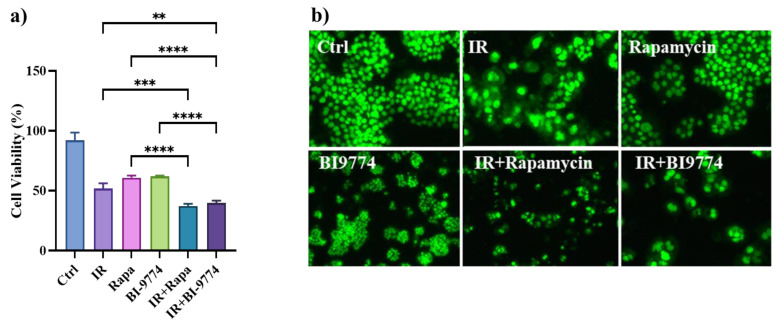
Panel (**a**): Cy-Quant viability assay in HT500 cells treated as described. In the combined treatments, cells were pre-irradiated (10 Gy), then incubated with Rapamycin (10 μM) or BI774(10 μM) for 96 h (mean of two experiments ± s.d.). Symbols indicate significant differences: ** *p* < 0.01; *** *p* < 0.001, **** *p* < 0.0001 (One-way ANOVA). Panel (**b**): micrographs of representative fields of cells stained with Cy-Quant fluorescent nuclear dye (200× magnification).

**Figure 8 cancers-15-02660-f008:**
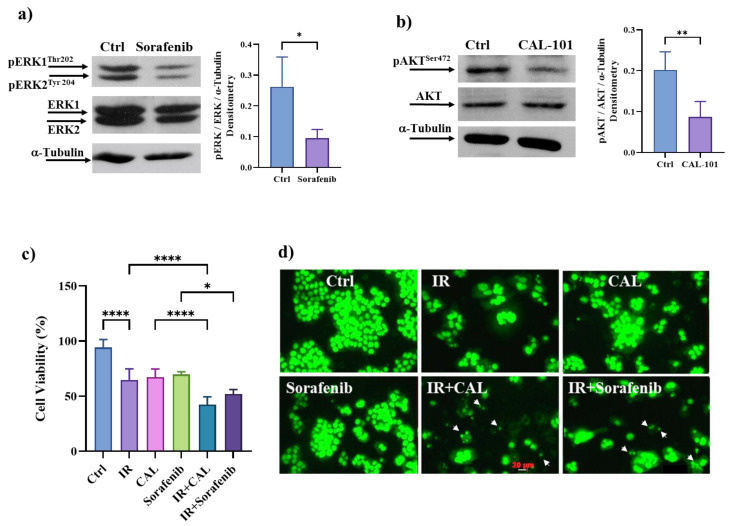
Panels (**a**,**b**): Western blot analysis of pERK1/2-ERK and pAKT-AKT. Cells were incubated for 20 min with Sorafenib (20 μM). or CAL-101 (10 μM). Band intensities were quantified by measuring optical density on Gel Doc 2000 and analyzed by Multi-Analyst Software; values in the graphs indicate protein expression normalized with respect to α-Tubulin (±s.d.) Symbols indicate significant differences: * *p* < 0.05, ** *p* < 0.01 Ctrl vs. Sorafenib or CAL-101, respectively (Student’s *t*-test). Panel (**c**): Cy-Quant cell viability assay in HT500 cells treated as reported. In the combined treatments, cells were pre-irradiated (10 Gy), then incubated with Sorafenib (20 μM) or CAL-101 (10 μM) for 72 h (mean of two experiments in quadruplicate ± s.d.). Symbols indicate significant differences: ** *p* < 0.01; **** *p* < 0.0001 (One-way ANOVA). Panel (**d**): Micrographs of representative fields of cells stained with Cy-Quant fluorescent dye (200× magnification). White arrowheads in panels IR + CAL and IR + Sorafenib indicate apoptotic bodies.

**Figure 9 cancers-15-02660-f009:**
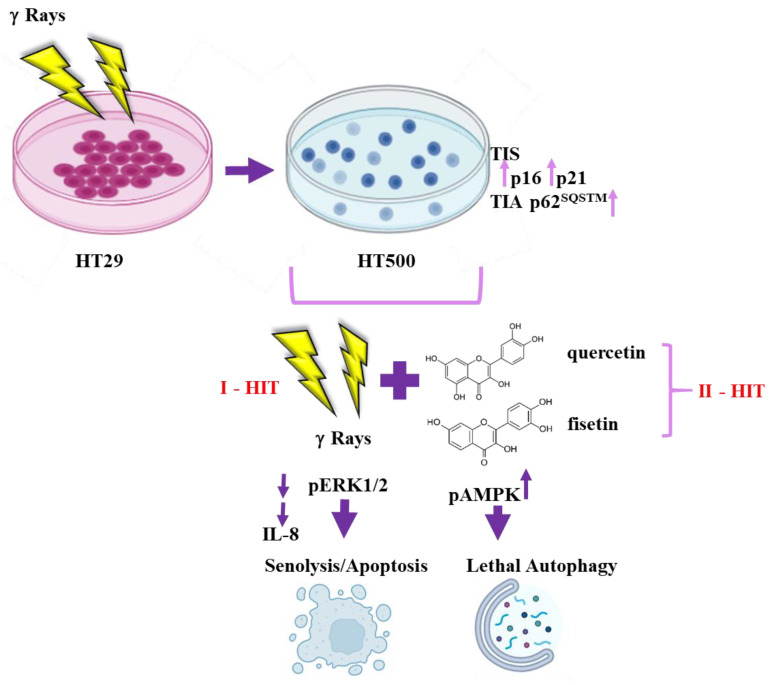
Scheme summarizing the molecular targets of Q and F in radioresistant HT500 cells.

**Table 1 cancers-15-02660-t001:** Cellular and biochemical characteristics of HT29 cells compared to their radioresistant derivative HT500.

Cellular/Biochemical Markers	HT29	HT500
EC_50_ IR (72 h) ^a^	21 Gy	58 Gy ***
SA-βGAL basal activity ^b^	1716.8 ± 8.6	3540 ± 59 ***
Fisetin/Quercetin radio-sensitizing effect ^c^	Not present	Synergic
(%) Intracellular peroxide after IR (5 min) ^d^	>226% ± 92	>119% ± 10 ***
(%) Intracellular GSH after IR (2 h) ^e^	No increase	>24% ± 5 **

^a^ Calculated with Crystal Violet Assay as described in [37]. *** *p* < 0.001 HT29 vs. HT500, Student’s *t*-test. ^b^ Expressed as Fluorescence (a.u)/μg total proteins as described in Materials and Methods. *** *p* < 0.001 HT29 vs. HT500, Student’s *t*-test. ^c^ Calculated with Combination Index Analysis (Compusyn software version 1.0) as described in [37] and Appendix A. ^d^ Measured with CM-DCFDA peroxide probe and expressed as percentage intracellular fluorescence compared to untreated cells as described [37]. *** *p* < 0.001 HT29 vs. HT500, Student’s *t*-test. ^e^ Measured with monochlorobimane intracellular probes and expressed as percentage intracellular fluorescence compared to untreated cells as described. ** *p* < 0.01 Ctrl HT500 vs. IR HT500, Student’s *t*-test.

## Data Availability

The data presented in this study are available on request from the corresponding author.

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
