# Peer review of "Senolytic Flavonoids Enhance Type-I and Type-II Cell Death in Human Radioresistant Colon Cancer Cells through AMPK/MAPK Pathway"

_cancers, 2023, doi:10.3390/cancers15092660_

Round 1

Reviewer 1 Report

The manuscript present interesting results about promising effects of quercetin and fisetin in reducing cancer cell resistance to ionizing radiations.

It requires minor revision before it can be accepted for publication.

1. Some sentences should be rewritten in order to facilitate the reader's comprehension. For example: LINE 48-50, PAGE 2: " …. the evolution of 'precision medicine' in oncology aims at bypassing the resistance to therapy and optimizing cancer treatments based on their biological age". What does the "biological age" refer to? LINE 58-60, PAGE 2: Through synthesis of metabolites, cytoplasmic constituents and organelles (generally defined cargo) are degraded in the lysosome to maintain cellular homeostasis [7]. What do "metabolites" refer to? LINE 71-72, PAGE 2: "It was initially believed that cellular senescence acted only as an autonomous tumor suppressor, but its complex functions throughout the life cycle are poorly understood." The sentence is unclear. LINE 150-154, PAGE 4: The description of the treatment protocol is not clear and should be improved. This comment applies also to other paragraphs of the MATERIAL AND METHODS section.

2. The description of previously published HT500 generation should be omitted from the RESULTS (paragraph 3.1).

3. In several figures the measurement units of the y axis are lacking.

4. In several figures the way of using symbols to indicate the p-values of the statistical comparisons is confusing

5. Reference number 22 should be properly formatted.

6. Minor English revisions are recommended. For example: - Supplementary figure 1, Y axis: "Cell viavility" instead of "Cell viability"; -Supplementary figure 1, legend: "... treated 96 h" instead of "... treated for 96 h"

Minor English revisions are recommended as indicated in the previous section.

Author Response

Reviewer #1 (R1)

The manuscript presents interesting results about promising effects of quercetin and fisetin in reducing cancer cell resistance to ionizing radiations.

It requires minor revision before it can be accepted for publication.

Reply. We warmly thank R1 for his/her positive evaluation of our work.

Query 1: Some sentences should be rewritten in order to facilitate the reader's comprehension. For example LINE 48-50, PAGE 2: " …. the evolution of 'precision medicine' in oncology aims at bypassing the resistance to therapy and optimizing cancer treatments based on their biological age". What does the "biological age" refer to? 1. Some sentences should be rewritten in order to facilitate the reader's comprehension. For example LINE 48-50, PAGE 2: " …. the evolution of 'precision medicine' in oncology aims at bypassing the resistance to therapy and optimizing cancer treatments based on their biological age". What does the "biological age" refer to? LINE 58-60, PAGE 2: Through synthesis of metabolites, cytoplasmic constituents and organelles (generally defined cargo) are degraded in the lysosome to maintain cellular homeostasis [7]. What do "metabolites" refer to? LINE 71-72, PAGE 2: "It was initially believed that cellular senescence acted only as an autonomous tumor suppressor, but its complex functions throughout the life cycle are poorly understood." The sentence is unclear. LINE 150-154, PAGE 4: The description of the treatment protocol is not clear and should be improved. This comment applies also to other paragraphs of the MATERIAL AND METHODS section.

Reply 1: In the revised version of the manuscript, we improved the explanation of the introduction following the indications received in his/her comments. The relevant sentence has been rephrased to increase clarity (see lines 51-54;62-66;78-81;150-159;164-166;168-175;219-227 in the revised manuscript).

Query 2: The description of the previously published HT500 generation should be omitted from the RESULTS (paragraph 3.1).

Reply 2: In the revised manuscript, we eliminated lines 262-270 of the original submission.

Queries 3-4: In several figures, the measurement units of the y axis are lacking. In several figures the way of using symbols to indicate the p-values of the statistical comparisons is confusing.

Replies 3-4: We are sorry for this carelessness probably due to the pressure to submit the manuscript on time. Since also the second reviewer requested the revision of the graphs, we re-elaborated all the figures with GraphPad Prism and utilized the ANOVA test to calculate the p-value resulting from multiple comparisons. The figure legends have been modified and corrected accordingly.

Query 5: Reference number 22 should be properly formatted.

Reply 5: In the original version, reference 22 was a duplicate of reference 24. We are sorry for the mistake. The error was corrected and the reference list was updated considering the addition of a few new references.

Query 6: Minor English revisions are recommended. For example: - Supplementary figure 1, Y axis: "Cell viavility" instead of "Cell viability"; -Supplementary figure 1, legend: "... treated 96 h" instead of "... treated for 96 h"

Reply-6: As reported above, we operated a substantial editing of the English style and grammar within the text to facilitate the reader's comprehension.

Reviewer 2 Report

Russo et al, showed that quercentin and fisetin, natural senolytics, with IR activates apoptosis and autophagy.
This work is really interesting and original, although there are still some important details that requiere improvement.

1- In Figure 1, 3, 4, 5, 6,7,8 the authors just showed two experiments and they use Student´s T test for the following comparison:

-control vs IR

-control vs Q alone

-control vs F alone

-Q vs IR+Q

-F vs IR+F

For this type of multiple comparison with more than two variables, Student´s T Test is not an acceptable statistical analysis. I strongly encourage the authors to use the correct analysis as ANOVA to see if the statistical difference is maintained.

2- In Figure 5e, they showed a Western Blot. In the control condition, it can be seen that the transfer has not ocurred properly in the middle of the band  affecting the sign. The authors use this sign (control) to compared the others conditions. As this is an important part of the message, I encourage the authors to repeat this Western Blott and measured again the signal showing  the new corresponding graphs.

The use of English is appropiate.

Author Response

Russo et al, showed that quercetin and fisetin, natural senolytics, with IR activates apoptosis and autophagy. This work is really interesting and original, although there are still some important details that require improvement.

Reply. We warmly thank R1 for his/her positive evaluation of our work.

Query 1: In Figure 1, 3, 4, 5, 6,7,8 the authors just showed two experiments and they use Student´s T test for the following comparison:
-control vs IR
-control vs Q alone
-control vs F alone
-Q vs IR+Q
-F vs IR+F
For this type of multiple comparison with more than two variables, Student´s T Test is not an acceptable statistical analysis. I strongly encourage the authors to use the correct analysis as ANOVA to see if the statistical difference is maintained.

Reply 1: The authors greatly appreciate this observation. To improve the quality and significance of our data, we re-elaborated all data shown in Figures 1-8 using GraphPad Prism software utilizing ANOVA test to calculate the p-values resulting from multiple comparisons (Figures 1, 3, 4, 7, 8). The slight changes in the values reported in Figure 1a are due to new data that have been added to the statistical analysis. The figure legends have been modified and corrected accordingly.

Query 2: In Figure 5e, they showed a Western Blot. In the control condition, it can be seen that the transfer has not occurred properly in the middle of the band affecting the sign. The authors use this sign (control) to compare the other conditions. As this is an important part of the message, I encourage the authors to repeat this Western Blot and measured again the signal showing the new corresponding graphs.

Reply 2: We agree with this observation. We substituted the immunoblot with the image deriving from a different, independent experiment where the same samples were analyzed.

Query 3: The use of English is appropriate.

Reply 3: We thank the reviewer for this comment. As stated above, English was further improved.
